

# Heatwave-like events affect drone production and brood-care behaviour in bumblebees

Yanet Sepúlveda, Elizabeth Nicholls, Wiebke Schuett and Dave Goulson

School of Life Sciences, University of Sussex, Falmer, Brighton, United Kingdom

## ABSTRACT

Climate change is currently considered one of the major threats to biodiversity and is associated with an increase in the frequency and intensity of extreme weather events, such as heatwaves. Heatwaves create acutely stressful conditions that may lead to disruption in the performance and survival of ecologically and economically important organisms, such as insect pollinators. In this study, we investigated the impact of simulated heatwaves on the performance of queenless microcolonies of *Bombus terrestris audax* under laboratory conditions. Our results indicate that heatwaves can have significant impacts on bumblebee performance. However, contrary to our expectations, exposure to heatwaves did not affect survival. Exposure to a mild 5-day heatwave (30–32 °C) resulted in increased offspring production compared to those exposed to an extreme heatwave (34–36 °C) and to the control group (24 °C). We also found that brood-care behaviours were impacted by the magnitude of the heatwave. Wing fanning occurred occasionally at temperatures of 30–32 °C, whereas at 34–36 °C the proportion of workers engaged in this thermoregulatory behaviour increased significantly. Our results provide insights into the effects of heatwaves on bumblebee colony performance and underscore the use of microcolonies as a valuable tool for studying the effects of extreme weather events. Future research, especially field-based studies replicating natural foraging conditions, is crucial to complement laboratory-based studies to comprehend how heatwaves compromise the performance of pollinators. Such studies may potentially help to identify those species more resilient to climate change, as well as those that are most vulnerable.

Corresponding author
Yanet Sepúlveda, ys321@sussex.ac.uk

# INTRODUCTION

Global climate change is currently considered one of the major threats to biodiversity (*Leadley et al., 2010*), causing distributional shifts (*Kerr et al., 2015*), population declines (*Parmesan & Yohe, 2003*), and even pushing some species to the brink of extinction (*Thomas et al., 2004*). This complex phenomenon, however, is not limited to an increase in mean global temperatures, which are predicted to rise between 1.2 °C and 4.5 °C under very low and very high emissions scenarios (*Intergovernmental Panel on Climate Change (IPCC), 2023*), but is also associated with increased climatic variability and extreme

weather events, such as heatwaves (*Meehl & Tebaldi, 2004*; *Christidis, Jones & Stott, 2015*). Although the definition of a heatwave varies, the United Kingdom's Meteorological Office defines it as a period of at least three consecutive days of hot weather based on local climatological conditions, with thermal conditions recorded above given thresholds (*McCarthy, Armstrong & Armstrong, 2019*). These extreme weather events are predicted to become more frequent, intense, and long-lasting (*Meehl & Tebaldi, 2004*), potentially creating acutely stressful conditions that are too short to allow organisms to adapt. These temperature extremes can exert considerable stress on plant (*Breshears et al., 2021*) and insect populations, pushing them to and even beyond their adaptive limits (*Harvey et al., 2020*). This stress may result in phenological disruptions and the structural reorganisation of food webs (*Harvey et al., 2020*) and could lead to a potential disruption in the performance and survival of ecologically and economically important organisms, such as insect pollinators (*Bridle & Vines, 2007*; *Huntley et al., 2010*).

Among insect pollinators, bumblebees (genus *Bombus*, family Apidae) are considered extremely valuable for their services to pollination (*Gunnarsson & Federsel, 2014*), particularly in the temperate northern hemisphere (*Woodard, 2017*). However, as cold-adapted insects, bumblebees might be affected by increasing temperatures associated to climate change (*Maebe et al., 2021*), particularly in areas where climatic conditions exceed species thermal tolerances (*Soroye, Newbold & Kerr, 2020*) or during prolonged heat events (*Rasmont & Iserbyt, 2012*). Previous research indicates that climate change is negatively affecting several bumblebee species, for example, by causing declines in some species because of their inability to shift their range (*Kerr et al., 2015*) or by forcing other species to move to higher altitudes in response to warming (*Pyke et al., 2016*). Heatwaves can affect bumblebees individually at the physiological level, for example when temperatures exceed their thermal threshold, and/or at a colony level, for example when extreme temperatures force workers to switch from foraging to colony thermoregulation (*Rasmont & Iserbyt, 2012*). The latter is particularly concerning because maintaining the nest at an optimal temperature is crucial for brood production, the development of the larvae and pupae, and the overall growth of the colony (*Weidenmüller et al., 2022*). Although, in general, bumblebees are capable of regulating the temperature of the colony and maintaining the nest temperature between 28 and 32 °C (*Vogt, 1986*; *Weidenmüller, Kleineidam & Tautz, 2002*), this comes at an energetic cost and unusually elevated ambient temperatures might cause thermal distress (*Vogt, 1986*) and affect bees' foraging performance and colony development (*Guiraud et al., 2021*).

Heatwaves generally occur during the summer, a period when some bumblebee species are actively producing drones (males) and gynes (*Goulson, 2003*). During one of these events, the brood might experience unusually elevated temperatures for a portion of their development (*Perl et al., 2022*), which can accelerate growth rate and decrease body size (*Davidowitz & Nijhout, 2004*), cause morphological changes, such as wing shape alterations (*Gerard et al., 2018*), and/or an impairment in adult behaviour (*Perl et al., 2022*). For example, *Gerard et al. (2018)* found that exposure to heatwave-like events (33 °C) during development can lead to smaller body sizes and wing asymmetry in *Bombus*

*terrestris*, which could lead to a reduction in pollination effectiveness (*Jauker, Speckmann & Wolters, 2016*). Bumblebees may also be exposed to heat stress during adulthood, which can impact essential behaviours, such as colony maintenance when bees increase fanning intensity (*Vogt, 1986*) and brood production, with more workers and queens produced at higher temperatures (*Holland & Bourke, 2015*). This exposure to high temperatures may also affect the number of sexuals produced and their reproductive capacity (*Guiraud et al., 2021*), which is essential for colony and population success.

Recent studies show that elevated temperatures can affect the reproductive fitness of bumblebees (*Campion, Rajamohan & Dillon, 2023*). In *Bombus ignitus*, an important pollinator in China and Japan, colony foundation, initiation, and male production has been shown to be negatively affected by temperature, with bees reared at 27 °C performing better than those at 30 °C (*Yoon, Kim & Kim, 2002*). In male bumblebees, high temperatures can have negative effects on fertility (*Martinet et al., 2021b*). For example, males of the cold-adapted species *B. jonellus* and *B. magnus* exhibited a significant decrease in sperm viability after being exposed to a simulated heatwave of 40 °C (*Martinet et al., 2021b*). These heatwave events need only occur once to affect an individual's responses, reduce fitness (*Kingsolver, Diamond & Buckley, 2013*; *Truebano et al., 2018*), and can lead to mortality in extreme cases (*Parmesan, 2006*; *Rasmont & Iserbyt, 2012*). *Rasmont & Iserbyt (2012)* reported that, after a record summer temperature of 33 °C in Lapland, Finland, bumblebee densities were abnormally low. How these extreme weather events affect colony survival and fitness, for example through increased worker mortality or sub-lethal effects on behaviour, requires more detailed investigation, both to increase our understanding of how heatwaves may contribute to bumblebee population declines, and how best to mitigate the effects of such events.

Here we investigated the impacts of simulated (5-day) heatwaves of two different magnitudes (30–32 and 34–36 °C) on the performance of queenless microcolonies of the bumblebee subspecies *Bombus terrestris audax*. Despite its heat tolerance and thermoregulatory capabilities, the colony performance of *B. terrestris*, particularly colony growth, has been reported to be affected by high temperatures (33 °C) (*Vanderplanck et al., 2019*). However, few studies have focused on assessing colony performance at heatwave temperatures equivalent to those that have been reported recently. In addition, microcolonies have been widely used to assess the impact of other stressors, such as pesticides (*Laycock et al., 2014*; *Dance, Botías & Goulson, 2017*) and are considered reliable predictors of the response of queenright colonies (*Tasei & Aupinel, 2008*). Queenless microcolonies are also an ecologically relevant scenario because queen mortality often occurs in field conditions (*Samuelson et al., 2018*) and can result in workers switching to egg laying to produce drones (*Duchateau, 1989*). To assess colony performance under simulated heatwaves, we measured and analysed parameters including (a) mortality of workers, (b) microcolony growth (production of drones), (c) discarded larvae, (d) food consumption, (e) worker weight changes, (f) drone body size (intertegular distance) and (g) dry body mass of drones. Furthermore, to determine whether heatwave events affect behaviours associated with colony maintenance and thermoregulation, we recorded (h) the number of workers engaged in wing fanning (flapping wings to cool the colony) and

(i) brood incubation (positioned on top of brood to keep them warm) during the 5-day heatwaves. We hypothesise that heatwaves of a higher magnitude would lead to a lower performance of bumblebee microcolonies, likely leading to higher mortality rates and decreased colony growth.

## MATERIALS AND METHODS

### Study organism

For this study, we selected *B. terrestris*, a widely distributed bumblebee species with a range that extends from Europe to North Africa, recognised for its high upper thermal tolerance (*Martinet et al., 2021a*; *Sepúlveda & Goulson, 2023*).

Four commercially reared colonies of *B. terrestris audax* were obtained from Biobest (Westerlo, Belgium) *via* Agralan Growers (Wiltshire, UK). Each colony contained ~50 female workers and a queen. Colonies were settled at standard ambient conditions (24 °C) (*Schmid-Hempel & Schmid-Hempel, 1998*) with *ad libitum* access to sucrose solution (Biogluc, Biobest *via* Agralan Growers) and pollen was provided every 48 h.

### Microcolony setup

The four colonies were divided into thirty queenless microcolonies by placing five bumblebee female workers from the same original colony in rectangular plastic boxes (L 16.8 cm × H 6.6 cm × W 11.6 cm) with a stainless-steel mesh lid for ventilation, a 10-ml syringe containing syrup (sucrose solution), and pollen feeders. Microcolonies were kept in a dark room with red light at ambient temperature (24 °C) and ~50% humidity (VicTsing Ultrasonic Humidifier) (*Gurel & Gosterit, 2008*).

Workers were individually tagged using Opalith ('Opalithplättchen') tags (EH Thorne (Beehives) Ltd, UK) to assess differences in weight change across treatments and to easily differentiate workers from drones. At the start of the experiment, each female worker was weighed to the nearest mg (VWR® LA Classic Analytical Balance). A small amount of wax from their colony of origin was provided in each microcolony to stimulate oviposition.

Microcolonies were subsequently assigned evenly to three treatments: two heatwave treatments and one control, with 10 microcolonies per group balanced across queenright colony origin. Before the heatwave simulation, one microcolony from the control group was removed from the experiment on day three, as all workers died during the first 2 days. This was potentially due to their failure to locate the food (sucrose solution) (feeder volume (10 ml) indicated that there was no consumption).

### Heatwave simulation

#### Selection of temperature treatments

*B. terrestris* colonies are generally located below-ground in old mammal nests, although they can also be established above-ground or on the ground surface (*Goulson, O'Connor & Park, 2018*). Studies indicate that the temperature within a bumblebee colony is subject to variation due to fluctuations in ambient temperature, direct sunlight exposure (*Richards, 1973*), and nest location (*Cumber, 1949*). However, data regarding air temperature within a bumblebee colony during a heatwave is limited. Research studies have found that the

ambient temperatures within underground nests can range from 20 to 34 °C (*Cumber, 1949*) and 13 to 34°C during 'normal' summer conditions (*Weidenmüller, Kleineidam & Tautz, 2002*); but this temperature range can be affected by changes in the external air temperature (*Richards, 1973*). In recent years, some UK regions where this species is found, have recorded temperatures exceeding 40 °C (*Kendon et al., 2023*). However, there are insufficient data regarding nest temperatures specifically during heatwave events. In this study, the conditions of the heatwave treatments were based on predicted ambient temperatures from the UKCP/Met Office Climate Projects. Microcolonies from the control group were exposed to 24 °C, which is considered normal summer temperature (*Larcom, She & van Gevelt, 2019*) and standard laboratory conditions for commercial bumblebee colonies (*Schmid-Hempel & Schmid-Hempel, 1998*). Microcolonies from the mild simulated heatwave were exposed to temperatures of 30–32 °C, while microcolonies from the extreme heatwave treatment were exposed to 34–36 °C.

### Simulation of heatwaves

Simulated heatwaves were achieved by placing ceramic infrared heaters (AIICIOO 150 W; 220 V (75 × 105 mm)) ~50 cm above the microcolonies to simulate heat coming from solar radiation. The number of infrared heaters depended on the magnitude of the heatwave. For the mild heatwave, temperatures ranged from 30 °C to 32 °C throughout the experimental heatwave, while for the extreme heatwave the temperatures ranged from 34 °C to 36 °C. Temperature was monitored with a data logger (Elitech GSP-6; Elitech (UK) Ltd, London, UK). Conditions were replicated for five consecutive days for 4 h (12:00–16:00) simulating the hottest part of the day.

Bumblebees were left to acclimate for 4 days prior to the heatwave simulation to initiate nest building without disruption. Once the acclimation period was over, the heatwave simulation was initiated. The heat lamps were turned on and the temperature increased gradually, reaching the selected temperatures after ~30 min. The temperature of each microcolony was monitored and recorded every 30 min using a data logger (Elitech GSP-6) to ensure temperature remained within the selected range for each treatment. Humidity levels remained at a constant 50–55% to reduce the likelihood of desiccation. After the heatwave simulation, the heat lamps were turned off and the temperature decreased gradually (~60 min) to ambient (~24 °C). The duration required to reach the target temperature and the subsequent return to ambient temperature after the heatwave simulation were excluded from our observations.

### Colony performance

Colonies were observed on a daily basis (during the 5-day heatwave treatments and during the following 4 weeks after the heatwave ended) and the following performance parameters were recorded: (a) daily mortality of workers over the 5-week period (number of dead workers per microcolony), (b) microcolony growth (production of drones at the end of the 5-week period), (c) the daily number of discarded larvae per microcolony, (d) food consumption (the volume of syrup feeders was taken every 3 days over the 5-week period and the total average was obtained (nine measurements per microcolony); ml), (e) weight

change of workers was obtained by collecting each individual bee with a marking cage (with the weight of the cage subtracted) and subsequently weighing them individually once a week over a 5-week period using an analytical balance (body mass; mg), (f) the body size (intertegular distance; mm) was obtained by measuring the distance between the bases of the wings on the bee thorax with a digital calliper after the 5-week period, and (g) the dry body mass of drones (dried at 60 °C for five consecutive days after the 5-week period and weighed to the nearest mg (VWR® LA Classic Analytical Balance)).

To determine if elevated temperatures alter brood-care behaviours associated with colony maintenance, bumblebee workers were observed during the simulated heatwave. We recorded (h) the total number of workers engaged in wing fanning (fanning with spread wings for at least 5 s (*Westhus et al., 2013*) and (i) the total number of workers incubating the brood (positioned on top of brood with curled abdomen for at least 5 s). The behaviour (h-i) of workers in each microcolony was observed continuously for 7 min divided into five intervals (1 min 40 s each) on a daily basis during the heatwave simulation for a total of 5 days (total of 35 min per microcolony). The maximum response (maximum number of workers fanning/incubating per interval) was recorded (adapted from *Weidenmüller, Kleineidam & Tautz, 2002*). For analyses, the values were averaged across the five intervals to obtain a single value per microcolony for each day.

## Statistical analyses

All statistical analyses were carried out using R version 4.1.2 (*R Core Team, 2021*). To test for potential influences of the temperature treatments, we ran several Generalised Linear Mixed Effects Models (GLMM) with either Binomial (for binary variables), Gamma (non-normal positive and continuous data), Negative binomial (for count data with overdispersion), or Poisson error structure (for count variables) and Linear Mixed Models (for normal data) using the lme4 package (*Bates et al., 2015*). All models included Microcolony nested within Colony of Origin as random effects to account for non-independent repeated measures and Treatment (three levels: control, mild heatwave, and extreme heatwave) as fixed term. Response variables included Status of individual worker after 5-weeks (binary variable; dead = 0 or alive = 1), Number of Drones Produced (Poisson error structure), Number of Discarded Larvae (Poisson error structure), Food Consumption (volume) (normal error structure), Weight Changes (mg) (normal error structure), Body Size (mm) and Body Mass (mg) (Gamma error structure), Number of Workers Fanning and Number of Workers Incubating (Negative binomial error structure). In addressing missing data attributed to bee mortality in the Weight Changes variable, we applied predictive mean matching (pmm) imputation using the mice package (*van Buuren & Groothuis-Oudshoorn, 2011*). The model with Number of Discarded Larvae as response also included Number of Drones produced per microcolony as additional fixed term to account for potential variations in microcolony size. *Post-hoc* estimated marginal means (EMMs) *post-hoc* tests were used accordingly to perform comparisons between groups using the emmeans() function in the emmeans package (*Lenth, 2023*). All plots were created using the ggplot2 package (*Wickham, 2016*). They are all based on raw data with the exception of the mortality plot, which was generated using the model's output data.

## RESULTS

### Mortality of workers

There were no statistically significant differences in bumblebee mortality across treatments (GLMM; $\chi^2 = 2.95$, df = 2, $p = 0.22$). However, bumblebees exposed to the extreme heatwave exhibited a slightly higher mortality in relation to those from the control group (24 °C) (GLMM with emmeans *post-hoc* test; $p = 0.24$) and the mild heatwave (32–34 °C) (GLMM with emmeans *post-hoc* test; $p = 0.38$) (Fig. 1). This led to the loss of two whole microcolonies from the extreme heatwave by the end of the study period.

### Colony growth

#### Drone production

The production of males (drones) differed among treatments (GLMM; $\chi^2 = 8.20$, df = 2, $p = 0.01$). Microcolonies exposed to the mild (30–32 °C) heatwave produced significantly more drones in comparison to the control group (24 °C) (GLMM with emmeans *post-hoc* test; $p = 0.01$), but not to the extreme heatwave (34–36 °C) (GLMM with emmeans *post-hoc* test; $p = 0.72$). Microcolonies subjected to the extreme heatwave exhibited higher drone production compared to the control group, although this difference was not statistically significant (GLMM with emmeans *post-hoc* test; $p = 0.09$) (Fig. 2).

#### Discarded larvae

During the study period, bumblebee workers from all treatments were observed to remove and discard larvae. However, there were no significant effects of either treatment (GLMM; $\chi^2 = 5.52$, df = 2, $p = 0.06$) or the number of drones produced per microcolony (GLMM; $\chi^2 = 0.51$, df = 1, $p = 0.47$). Figure is available in Supplemental Material (Fig. S1).

### Food consumption and weight changes

Over the 5-week study period, all microcolonies consumed similar volumes of sucrose solution (mean ± SE; control = 6.59 ml ± 0.22; mild heatwave = 6.37 ml ± 0.23; extreme heatwave = 6.25 ml ± 0.27). The total amount of volume of consumed sucrose solution was not significantly affected by heatwave exposure (GLMM; $\chi^2 = 0.72$, df = 2, $p = 0.69$).

Bumblebees from all treatment groups lost weight over the 4-week period. However, these differences were not statistically significant (LMM; $\chi^2 = 3.00$, df = 2, $p = 0.22$). On average, bumblebees in the control group lost 33.1 ± 10.1 mg (mean ± SE) from the start to the end of the experiment. Meanwhile, bumblebees exposed to the mild heatwave and extreme heatwave lost, on average, 63.6 ± 11.0 mg and 56.3 ± 8.78 mg (mean ± SE), respectively. Figures are available in Supplemental Material (Figs. S2 and S3).

### Drone body size and dry mass

There were no significant differences in the body size (ITD; mm) (GLMM; $\chi^2 = 2.96$, df = 2, $p = 0.22$) or the dry body mass (mg) (GLMM; $\chi^2 = 0.21$, df = 2, $p = 0.89$) of adult drones produced across treatments. Figures are available in Supplemental Material (Fig. S4).

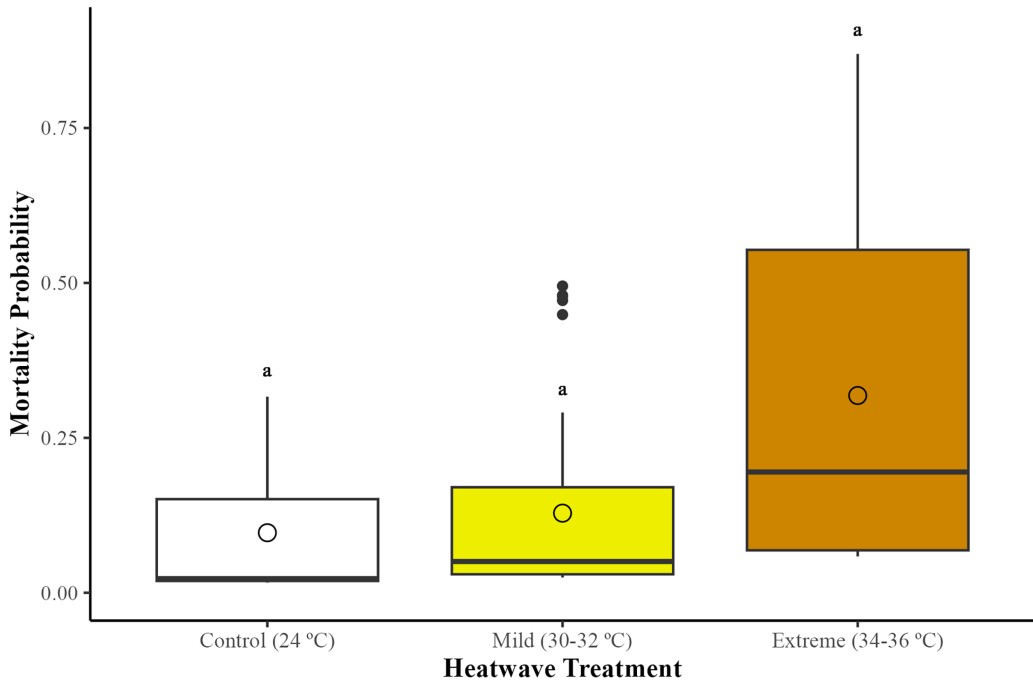

**Figure 1 Effects of temperature on worker mortality among treatments.** Mortality probability was assessed across 29 bumblebee microcolonies (*Bombus terrestris audax*) (Control = nine microcolonies, 41 bees; mild heatwave = 10 microcolonies, 47 bees; extreme heatwave = 10 microcolonies, 49 bees). The boxes represent the interquartile ranges, the horizontal lines within the boxes show the medians, and the circles indicate the means of the data points. Whiskers extend to 1.5 times the interquartile range from the first and third quartiles, displaying the range of the data. Any data points outside these whiskers are considered outliers and are represented as individual points.

## Brood-care behaviours: wing fanning and brood incubation

### Wing fanning

The incidence of wing fanning varied significantly between treatments (GLMM; $\chi^2 = 34.49$, df = 2, $p < 0.01$). Microcolonies exposed to the extreme heatwave exhibited a higher proportion of workers engaged in wing fanning in comparison to those exposed to the mild heatwave (GLMM with emmeans *post-hoc* test; $p < 0.01$) and the control group (24 °C) (GLMM with emmeans *post-hoc* test; $p < 0.01$) (Fig. 3). Workers exposed to the mild heatwave exhibited a slight increase in wing fanning behaviour; however, there were no significant differences when compared to the control group (GLMM with emmeans *post-hoc* test; $p = 0.35$).

### Brood incubation

There were no significant differences in the incidence of brood incubation between temperature treatments (GLMM; $\chi^2 = 4.42$, df = 2, $p = 0.10$).

## DISCUSSION

This study investigated the effects of heatwaves of different magnitude on the performance of bumblebee (*B. terrestris audax*) microcolonies. We found that exposure to a mild 5-day

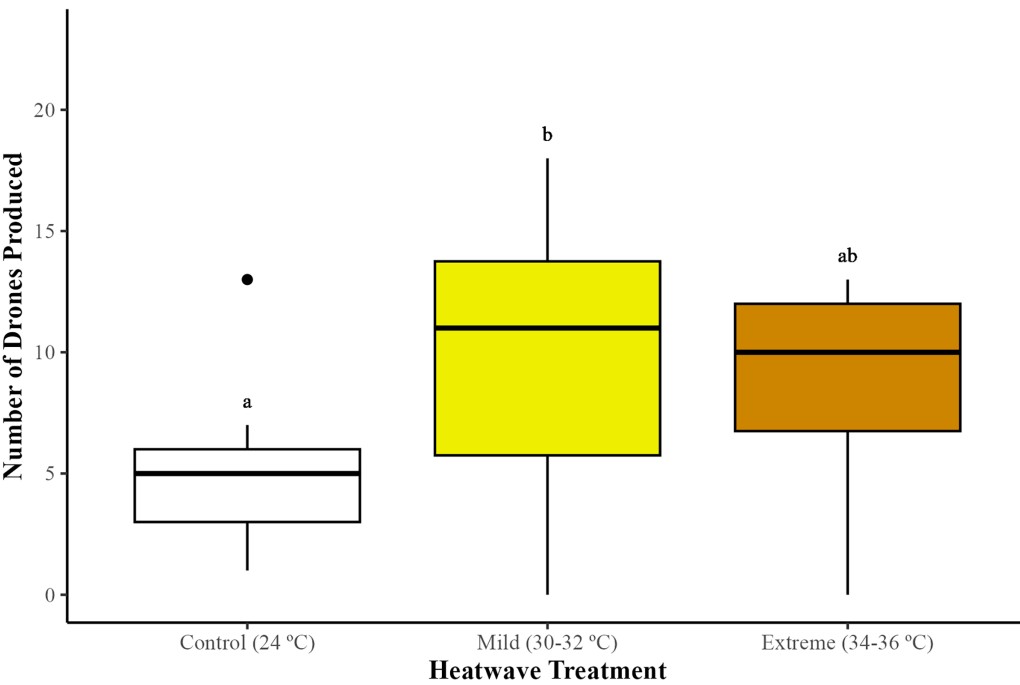

**Figure 2 Impact of heatwaves on the average number of males (drones) produced.** Production of drones was assessed across 29 bumblebee microcolonies (*Bombus terrestris audax*) (Control = nine microcolonies; mild heatwave = 10 microcolonies; extreme heatwave = 10 microcolonies). The boxes represent the interquartile ranges and the horizontal lines within the boxes show the medians. Whiskers extend to 1.5 times the interquartile range from the first and third quartiles, displaying the range of the data. Any data points outside these whiskers are considered outliers and are represented as individual points. Different letters indicate statistically significant differences ($p < 0.05$).

heatwave of 30–32 °C resulted in increased offspring production compared to those exposed to an extreme heatwave of 34–36 °C and to the control group (24 °C). Brood-care behaviours were also impacted by the magnitude of the heatwave. Wing fanning occurred occasionally at temperatures of 30–32 °C, whereas at 34–36 °C the proportion of workers engaged in this thermoregulatory behaviour increased significantly. However, we did not find evidence indicating that exposure to heatwaves affects the survival of workers or the body size and mass of drones.

Although our analyses did not reveal a significant effect of heatwave exposure on bumblebee mortality, microcolonies exposed to the extreme heatwave exhibited the loss of several workers and two whole microcolonies. This finding contrasts with previous research that has documented the effects of prolonged exposure to sublethal temperatures on bumblebee survival. Several studies have shown that sublethal temperatures can become lethal if they persist for an extended period of time (*Rasmont & Iserbyt, 2012*). For example, *Bombus impatiens* workers exposed to 36 °C for a prolonged period of time (5 h) experienced greater worker mortality compared to those exposed to 26 °C (*Quinlan et al., 2023*); while in *B. terrestris audax* workers, repetitive exposure to sub-lethal temperatures of ~50 °C led to high mortality and irreversible loss of coordination despite their high heat tolerance (*Sepúlveda & Goulson, 2023*).

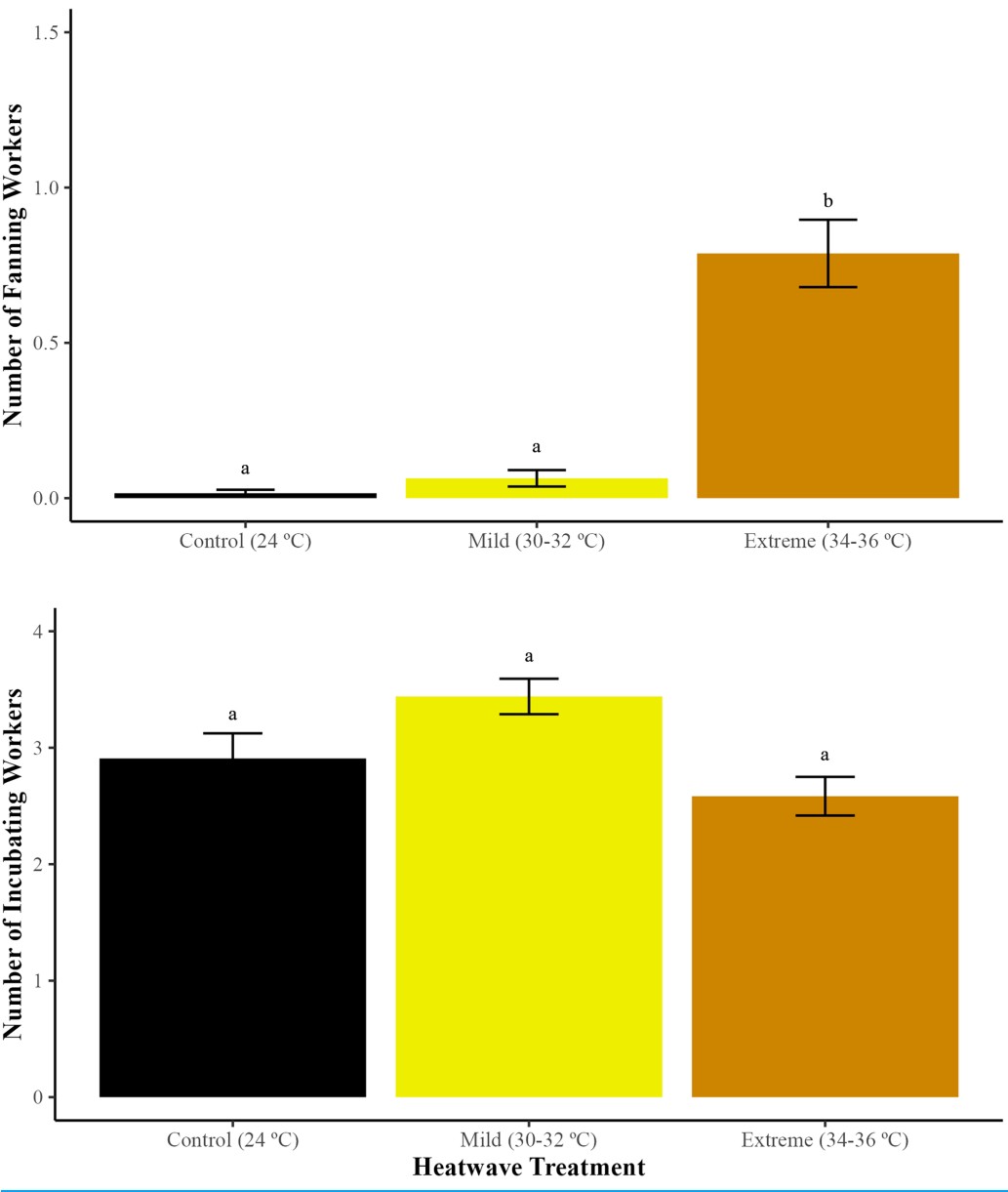

**Figure 3** **Average number of bumblebee workers engaged in wing fanning and incubating the brood during the 5-day heatwave.** The total number of workers engaged in wing fanning (fanning with spread wings for at least 5 s) and the total number of workers incubating the brood (positioned on top of brood with curled abdomen for at least 5 s) were recorded during heatwave exposure. Data represent mean values (SE) for the two treatments and one control group. Different letters indicate statistically significant differences ($p < 0.05$).

High mortality levels have also been observed in other heat-tolerant bees, such as *Tetragonula* stingless bees, which experienced 95% mortality at temperatures below their upper thermal limits after longer periods of heat exposure (*Nacko et al., 2023*). However, most of these studies use higher experimental temperatures, potentially accounting for the observed differences in mortality rates. For example, *Martinet et al. (2021a)* showed that *B. terrestris* can reach heat stupor when exposed to temperatures of 40 °C for a prolonged

time. The temperatures used in our study are also lower than those experienced during recent heatwave events. *B. terrestris* inhabits regions ranging from Europe to North Africa, where ambient temperatures exceeding these levels have already been recorded. For instance, during the 2022 summer heatwave in the UK, some regions recorded temperatures exceeding 40 °C (*Kendon et al., 2023*). Similarly, in 2021, the Mediterranean region (Italy) registered record-breaking temperatures of 48.8 °C, while 50.3 °C was recorded in North Africa (Tunisia) (*United Nations Climate Change, 2021*). It is essential to note, however, that temperatures within the nest, especially for species that establish nests underground, may not reach such extremes. While nests that are located above-ground can experience high thermal extremes, those located below-ground may exhibit relatively stable temperatures due to insulation (*Mullan, 2022*). Thus, it remains necessary to investigate the nest temperatures during extreme weather events. Finally, it is also plausible that our sample sizes, *i.e.*, number of colonies, might have influenced the detection of a significant difference in mortality levels. Future studies focused on the effect of elevated temperatures on bumblebees and other insects should include larger sample sizes to enhance the statistical power and robustness of the findings.

Interestingly, bumblebees exposed to the mild heatwave produced significantly more drones in comparison to the extreme heatwave and the control group. Previous research indicates that *B. terrestris* need to keep the nest at constant temperatures between 28 °C and 32 °C (*Vogt, 1986*), which could explain the higher drone production on those exposed to temperatures of 30–32 °C. Temperatures of this range are considered optimal for the brood in bumblebee colonies and microcolonies (*Livesey et al., 2019*) under both field (*Weidenmüller, Kleineidam & Tautz, 2002*) and laboratory conditions (*Vogt, 1986*). Thus, nests kept at this temperature range might therefore result in a more efficient colony development due to a decrease in the energy spent on thermoregulation (*Vollet-Neto, Menezes & Imperatriz-Fonseca, 2011*; *Wynants et al., 2021*). Recent studies (*Guiraud et al., 2021*) have shown that both worker and drone production can remain constant even at high, stressful temperatures (33 °C). For example, *Amin, Suh & Kwon (2008)* found that *B. terrestris* queens that had hibernated for 4 months and had been activated (termination of diapause) at 36 °C produced the highest number of drones in comparison to those activated at 24 °C, 28 °C, and 32 °C. This low sensitivity to elevated temperatures could explain the species' capacity to thrive in a broader range of climatic zones (*Martinet et al., 2021a*). On the other hand, recent studies focused on bumblebees and honeybees suggest that higher queen and/or worker production may represent an emergency-state that leads to more investment in reproduction for the ultimate success of the colony during stressful conditions (*Schott et al., 2021*; *Guiraud et al., 2021*). However, further experimentation would be required to comprehend whether higher drone production may be a stress response that enables bumblebees to ensure future colony foundation through future queen fertilisation and subsequent nest foundress activity by new queens (*Belsky, Camp & Lehmann, 2020*).

Previous work on insects has shown that high developmental temperature can lead to decreased body size (*Atkinson, 1994*). For example, the body size of high-Arctic butterflies has decreased as a response to high summer temperatures (*Bowden et al., 2015*), while the

decrease in size of some large-bodied beetles, such as *Scaphinotus angusticollis*, has been correlated with increased temperatures (*Tseng et al., 2018*). However, we did not find any significant differences in the body size or body mass of bumblebee drones across treatments. This is consistent with *Guiraud et al. (2021)*, who found that colonies of *B. t. audax* produced smaller workers, but not drones, when reared at temperatures of 33 °C. This indicates that effects of temperature on body size might be sex-dependent, as it is suggested that the body size of each sex may respond differently to temperature or have no response at all (*Fenberg et al., 2016*). Furthermore, it is important to note that other factors, such as nutrition, can play a role in body size in bees (*Nicholls, Rossi & Niven, 2021*). For example, honeybee workers reared on high protein diets were smaller and lighter on emergence compared with those reared on high carbohydrate larval diets (*Nicholls, Rossi & Niven, 2021*); while in *B. terrestris*, queens produced smaller drones in poor environments with reduced food availability (*Schmid-Hempel & Schmid-Hempel, 1998*). Similarly, tolerance to heat stress can also be influenced by nutrition. For example, stingless bees (*Melipona subnitida*) with previous access to food showed higher critical and lethal temperatures in comparison to unfed bees (*Maia-Silva et al., 2021*). In our study, microcolonies received a constant supply of syrup and pollen and there were no significant differences in syrup consumption among treatments. As this may not accurately reflect field conditions where bumblebees need to actively gather food resources, further research is necessary to explore how bumblebee behaviours and responses to heat stress are affected during natural heatwaves. Thus, it remains important to continue exploring the interactive effects of nutrition and elevated temperature on body size and other morphological traits, as any changes could directly affect foraging (*Greenleaf et al., 2007*) and, consequently, colony development (*Gérard et al., 2022*).

We also found that brood-care behaviours, particularly wing fanning, were impacted by the magnitude of the heatwave. Wing fanning occurred occasionally at temperatures of 30–32 °C, whereas at 34–36 °C the proportion of workers engaged in this thermoregulatory behaviour increased significantly. This is consistent with *Vogt (1986)*, who observed a higher occurrence of wing fanning and low levels of brood maintenance in colonies of *B. impatiens* and *B. affinus* exposed to 33–39 °C in comparison to 28–32 °C. Similarly, an increase in fanning accompanied by a reduction in incubation as a response to increased temperatures was observed in *B. bifarius nearcticus* workers (*O'Donnell & Foster, 2001*). Wing fanning is an effective strategy to cool the colony and reduce the levels of respiratory gases ($CO_2$) within a nest (*Weidenmüller, 2004*). However, increasing the recruitment of workers for thermoregulation at the expense of other essential behaviours in response to extreme temperatures may affect the growth of colonies (*Vogt, 1986*). For example, at temperatures of ~32 °C, *B. terrestris* workers decrease their foraging activity by almost 70% compared to ~25°C (*Kwon & Saeed, 2003*). Reductions in foraging and other essential behaviours can result in significant losses at a larger scale and even population collapses (*Zeaiter & Myerscough, 2020*). Therefore, investigating the effects of heatwaves in queenright colonies when foraging is required remains essential. Finally, when assessing impacts of temperature on colony performance, the number of workers in a colony is important to consider as the size of a colony can help buffer environmental

stress (*Vanderplanck et al., 2019*). Small colonies, such as those from our experiments, might be more sensitive to extreme temperatures (*Weidenmüller, Kleineidam & Tautz, 2002*), since they cannot recruit more workers to carry out essential tasks and generally require the presence of other members to initiate fanning (*Cook & Breed, 2013*). While studies have indicated that small microcolonies may be more successful at brood incubation given that additional workers consume resources without a corresponding increase in performance (*Livesey et al., 2019*), it is crucial to investigate whether this phenomenon extends to wing fanning and wild small colonies as well. Therefore, decreases in colony size as a result of heat exposure and/or its interaction with other stressors should be considered when predicting responses to extreme weather events (*Vanderplanck et al., 2019*).

## CONCLUSIONS

In the present study, we demonstrate that heatwaves can have significant impacts on the performance of *Bombus terrestris audax*. Unexpectedly, bumblebees exposed to the mild heatwave produced more drones in comparison to those exposed to the extreme heatwave and the control group. We also found that brood-care behaviours were impacted by the magnitude of the heatwave, with higher temperatures leading to more wing fanning. Changes in behavioural responses to elevated temperatures can affect colony growth and performance, especially in small colonies with limited number of workers that can invest in other essential tasks.

Our findings add to an existing body of work on the impact of extreme weather events on bees. However, further experiments are necessary to understand the impact of elevated temperatures on different developmental stages and to assess the long-term effects of heat stress and the potential behavioural responses that might enable them to avoid and/or cope with extreme temperatures. It is important to highlight that, in our study, we provided a constant supply of syrup and pollen. Therefore, further research is necessary to explore how bumblebee behaviours are affected when foraging is required during these events. Moreover, even though *B. terrestris* is considered a heat-resistant species, the negative impacts of extreme temperatures have been documented. Thus, we strongly encourage exploring the effects of heatwaves of comparable magnitude to those observed in recent years and to include larger sample sizes to enhance power and robustness. Finally, we suggest carrying out more research to comprehend the impact of heatwaves on other bee species that might be more sensitive to temperature extremes to predict and prevent population declines of pollinators in a warming world.

### Funding

This research was supported by Consejo Nacional de Humanidades, Ciencias y Tecnologías (CONAHCYT) through a doctorate scholarship. The funders had no role in study design, data collection and analysis, decision to publish, or preparation of the manuscript.

### Grant Disclosures

The following grant information was disclosed by the authors:
Consejo Nacional de Humanidades, Ciencias y Tecnologías (CONAHCYT).

### Competing Interests

The authors declare that they have no competing interests.

### Author Contributions

- Yanet Sepúlveda conceived and designed the experiments, performed the experiments, analyzed the data, prepared figures and/or tables, authored or reviewed drafts of the article, and approved the final draft.
- Elizabeth Nicholls conceived and designed the experiments, authored or reviewed drafts of the article, co-supervision, and approved the final draft.
- Wiebke Schuett conceived and designed the experiments, authored or reviewed drafts of the article, co-supervision, and approved the final draft.
- Dave Goulson conceived and designed the experiments, authored or reviewed drafts of the article, supervision, and approved the final draft.

### Data Availability

The raw data and R codes are available at Mendeley Data: Sepulveda, Yanet (2024), "Raw Data and RCode: Heatwave-like events affect drone production and brood-care behaviour in bumblebees", Mendeley Data, V3, DOI 10.17632/ytyyjcz3w3.3.

### Supplemental Information

Supplemental information for this article can be found online at http://dx.doi.org/10.7717/peerj.17135#supplemental-information.

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
