# Peer review of "Heatwave-like events affect drone production and brood-care behaviour in bumblebees"

_PeerJ, doi:10.7717/peerj.17135_

## Round 0.1 · original submission · Major Revisions

Please provide more information on the data provided. Please check on whether you can use a Poisson distribution on non-integer count-like data. Make figures and how you indicate significance consistent within the paper. Overall the reviewers have provided a lot of good feedback, which should be addressed as much as possible.

·

Basic reporting

All comments in attached PDF.

Experimental design

All comments in attached PDF.

Validity of the findings

All comments in attached PDF.

Additional comments

All comments in attached PDF.

·

Basic reporting

Overall, there is good background information in the introduction, and it is well-structured. I especially appreciated the tie in to how physiological effects at the individual level can affect the whole nest at the colony-level (specifically, the sentence beginning on line 92 through 95). When discussing previous studies, especially regarding thermal physiology research, it is important to keep the context of the original cited papers which I think the authors did a good job of doing by bringing up the specific temperatures that were used in previous studies (for example, line 109).

Professional English language was used throughout the paper. The sections were well-written with no grammar issues that would impede understanding of the research. The structure of the paper is standard and appears to use the format requested by PeerJ. The figures are very clear, straightforward, and properly referenced within the text.

Below are my major comments, the first is relevant to the introduction and the second is about the raw data/R script. Following these comments are minor comments relevant to the introduction.
I'd like to see just a little bit more of a direct connection between the background and how this research fits into the gap of knowledge. The introduction is fairly well done, but I'm not certain that I understand whether this has been done before in this species, or why it is so important that we study the colony performance of this species in particular. I think this connection is almost made, but not quite. I would just add a little bit more between the last two paragraphs of the introduction. This might also help explain the ecological or agricultural relevance of heatwaves to this specific species.

The raw data and R scripts were provided, and I was able to download them. The raw data is missing metadata or some description of the columns in the datasets (include as “read me” document or in a tab on the excel file provided). The R script is well-ordered and matches the paper layout. However, I could not successfully run the R script because there seemed to be some files missing. I would recommend downloading what was provided and running it as is to find these same areas that were troublesome for me. Specifically, here are some of the issues that I found:
- “new.data_2” is undefined
- emmeans should be included as one of the packages listed in the beginning to load in
- the summary_table from the drone production plot is undefined/not present
- There seems to be a difference between the Weight_data file provided and the Weight_Data_Change data that is used in the script. This may be important because it sounds like the data may have been transformed in some way.
- Drone size is a file that is undefined/not provided

As a final and more important note, on the behavior data, please check on whether you can use a Poisson distribution on non-integer count-like data. At the very least, I think the data should be transformed so they are not decimals because I worry that R is not properly running the data and coercing it to run based on the warnings it gave me. Please check this and defend using Poisson if this method is determined appropriate still.

Line 79 – Define what this source means by ‘hot’ weather. This seems somewhat vague, especially if this is the definition you want to use for a heatwave which is an important basis for this study.

Line 112-113 – I would be more specific here about how heat can affect colony maintenance and brood production behaviors.

Experimental design

The overall set up of the experiment is well-defined by the last paragraph of the introduction which explains the objective and hypothesis of the study. There are some aspects of the methodology that are confusing. I don’t think there are any major problems with the methods (and it was good to see colony of origin/relatedness taken into account in the statistical models to avoid pseudoreplication), but the methods need to be explained more clearly and properly justified where appropriate. I will list my comments in order of importance:

On line 215, when you begin to describe how everything was measured, it is not clear to me if colonies were observed and measured every day for 5 days during the heatwave treatment and then only once after 4 weeks or every day for the 4 weeks too? How many days was that 4 weeks- exactly 28 days or some other number of days? During the initial 5 days, when were they measured (i.e. during the heatwave or after the heatwave each day)? Were certain data recorded only during the 5 days and certain data recorded after 4 weeks? When was the drone dry weight taken- once after the heatwave or once after the 4 weeks? The volume of the feeders was measured every 3 days for a total of 9 times, so that is 27 measurements. Does that mean that the feeders were only measured after the heatwave during the 4 weeks following? Please make all of these more clear. I would also suggest briefly describing the methods for each in the figure legends of the graphs. Right now, it is not clear how many measures were taken for each metric and how that impacts the results that are shown.

On line 228, when you describe the behavioral data collection, it is also not entirely clear. Specifically, did you record via scan sampling after each 1 m 40 s interval like a snapshot sampling, or was it a continuous sampling for each interval where you recorded the highest number of workers exhibiting that behavior all at once.

I would highly recommend making a table of your statistical information. It can go in supplemental if needed, but I think it would be much easier to describe what was done for each type of data collected. I would set up the first column as each response variable that was measured and then have a columns for what statistical model was used, what was in that model (predictor and random variables used), and a stat summary could even be included here as well that can be referenced in the results section. (You could possibly also include how many days each variable was measured to help improve the clarity of what was measured and when as I mentioned in my first major comment). You have explained this section really well, but I think this will be even easier to quickly glance through and understand.
Please explain why the model with the number of discarded larvae also included the number of drones produced (line 248).

Other major comments by line number:
Line 154 – what is its upper thermal tolerance, or its full range of thermal tolerance? Any other thermal tolerance metrics that could be included about this species that aren’t mentioned elsewhere in the paper?
Line 156 – can you provide the rearing temperature of these colonies from Biobest? I think this is relevant information that could inform what the ‘natural’ thermal environment is for these reared colonies. As you stated in the introduction, rearing conditions during development can affect adult physiology and behavior.
Line 161 – I apologize if I missed it, but can you add some citations and discuss any previous research that has looked at the effects of splitting up a colony into microcolonies (i.e. the effects on behavior). It may be worth considering how manipulations like this can affect thermoregulation (which I think you may mention a little bit in the discussion around line 406 about the effect of colony size).
Line 164 – can you briefly describe the feeders more or cite something that describes them? I ask because you said that one colony possibly couldn’t find them and died as a result.
Line 165-166 – I would like to know more about the stability of the temperature and humidity for this experimental set-up. It sounds like an incubator was not used. You provided information about the range of humidity. Can you provide more details about the range of temperatures that were experienced?
Line 166 and 210 – Is there research to support that 50-55% RH are normal conditions within a nest of this species? If not, please specify.
Line 169 and 220 - How was the weight taken of these bees? Was it done via chilling? This needs to be specified because different methods would have impacts on thermal tolerance. Also, how often was each bee weighed? Was this daily, or how often were they measured?
Line 175 – did this control microcolony die during the acclimation period or during the heatwave?
Line 187 – I know that you said there wasn’t much information about the nest temperatures of this species, but what species are these citations using? The ranges are incredibly wide, and some of them encompass your low heatwave. I think because these bees are acclimated at 24 degrees that 32 degrees could still count as a heatwave even if it is within normal parameters of nest temperatures during certain times of the year, but it may be important to be more clear about this and defend why what you did still counts as a heatwave. My concern is that you used ambient temperature as your basis rather than a natural nest temperature (line 191). You’ve stated that nests can be thermoregulated, but then continue on to state that you pulled your treatment temperatures from ambient weather station data. This is worth explaining more.
Line 205 – why was a 4-day acclimation chosen? Is there research you can cite that discusses the cost and benefit of choosing different acclimation times?
Line 207 – Heat lamps were used to increase the temperature to simulate the heatwave, but how were you able to do a high and low heatwave with the lamps? Was the lamp closer to the bees to increase the heat or was an additional lamp used? How consistent were these lamps in the temperatures that they produced based on what you measured using the data logger?

Validity of the findings

The conclusions are overall well-stated and relevant to the results of this study. The background information provided in the discussion is substantial and helps to explain the findings of this study. As mentioned previously in the Basic Reporting section, I would check on the use of Poisson for the non-integer behavioral data. Aside from that, the data analysis is statistically sound. I do think there are some areas where care should be taken not to overstate nonsignificant data. Specifically, it may be a bit misleading on line 258 to state that, “Bumblebees exposed to the extreme heatwave exhibited higher mortality in relation to those from the control group (24 C) and the mild heatwave…” when this is actually non-significant as you say two sentences later. Again, on line 306, you say a slight increase when it is not statistically significant, so there actually is no difference. And again, on line 311, there is no significance. I would be very careful to be clear that there seem to be visual trends that are not statistically significant or supported by data analysis. As it stands, right now, it feels a bit like you are leading the reader to make certain assumptions. Additionally, if you are going to briefly summarize the results again in the conclusion section, I think it may still be important to mention your null results as well as your significant results. It is still interesting and important to highlight areas where species are resilient to changes in their thermal environment.

Some background information has been presented that concerns me about the rationale for choosing the temperatures of the study. I think because there was acclimation at 24 degrees, having a heatwave at 32 degrees may still be acceptable. But, as I mentioned before, I think citations are needed for why a 4-day acclimation period was chosen, and information is needed about what the rearing temperatures were of these colonies at Biobest. Additionally, I think the selection of temperatures used should be better explained early on. In this study, the conditions of the heatwave treatments were based on ambient weather station conditions rather than natural nest temperatures, for normal summer temperature (24 C), mild heatwave (30-32 C), and extreme heatwave (34-36 C) (line 190-195). It was mentioned in the introduction that there is little information about how internal nest temperature is affected by heatwaves, so it makes sense that it is hard to choose temperatures that are representative of real heatwave effects. However, there is also information stated early on about the natural temperature range of these nests that makes me question why 24 degrees C was chosen as the ‘normal’ temperature. For instance, line 98-99 states, “…in general, bumblebees are capable of regulating the temperature of the colony and maintaining the nest temperature between 28 and 32 ºC (Vogt, 1986; Weidenmüller, Kleineidam & Tautz, 2002).” And then again, line 345-346 states, “Previous research indicates that B. terrestris need to keep the nest at constant temperatures between 28 and 32 ºC (Vogt, 1986)” and that this is also optimal for brood. It was also mentioned that 24 C was standard ambient conditions according to Schmid-Hempel & Schmid-Hempel (1998). What makes this temperature standard in this study and why is that more standard that the more recent research that seems to conflict with that? I would try to frame this better, if possible.

The first paragraph of the discussion (line 316) I think is good in discussing differences in other studies that assessed thermal performance at temperatures below the upper thermal limit of a species. It may be worth mentioning with some of these studies what their acclimation/control temperatures were compared to this study because they may have tested the same or higher temperatures but acclimated the bees at a different temp as well. Later, the authors begin to discuss how the temperatures that they used are likely lower than predicted ambient temperatures for heat waves, and that their sample size needs to be higher. While this may be true, I don’t know if this is the most important thing to mention here. I would bring those points up later, but first I think it may be more important here to begin discussing the information that is brought up in the next paragraph (line 345-350). I would consider mentioning this information here and stating that this temperature range might actually be optimal for this species, and that 24 degrees C might be relatively cold. This may explain a lot of the nonsignificant effects that were found. These temperatures tested just might not have been stressful enough for this species even if they were acclimated at 24 degrees C.

For line 333-337, whenever discussing broad weather station climate data, I would be really careful and include caveats about how microclimatic data is very different than climatic data. As you have stated before, the nest microclimates can vary compared to the outside environment due to behavioral thermoregulation. Future studies could look at this more to really parse out these differences.

Minor comments by line number:
Line 266 – Please include the actual p values that were found rather than just p < 0.05. This comes up in multiple spots. I would make sure that somewhere the actual p values are stated.
Line 326 – what are these sublethal temperatures that were used previously for B. terrestris?
Line 329 – Rather than saying, “temperatures lower than their thermal limits.” I would clarify this by saying, “temperatures below their upper thermal limits.”
Line 331 - This sentence is not super useful here since the other examples you mention already have temperatures higher than what you tested: “For example, Martinet et al. (2021a) showed that B. terrestris can reach heat stupor when exposed to temperatures of 40 ºC for a prolonged time.” This sentence seems to be discussing more of a thermal tolerance study rather than thermal performance. This might be more useful when you first discuss this species as having a high thermal tolerance towards the beginning of the paper.
Line 341 – I would be clear here what you mean by larger sample size. With social insects, this could mean more colonies or more individuals per colony (or in this case, there is another level of microcolony making it even more complex).

Line 386 – this may or may not be a relevant citation that you want to include, but more studies are finding that providing a food source to insects while they undergo thermal tolerance assays will actually increase their ability to withstand higher and lower temperature extremes (Bujan & Kaspari, 2017; Macías-Macías et al., 2011; Maia-Silva et al., 2021).
- Bujan, J., & Kaspari, M. (2017). Nutrition modifies critical thermal maximum of a dominant canopy ant. Journal of Insect Physiology, 102, 1–6. https://doi.org/10.1016/j.jinsphys.2017.08.007
- Macías-Macías, J. O., Quezada-Euán, J. J. G., Contreras-Escareño, F., Tapia-Gonzalez, J. M., Moo-Valle, H., & Ayala, R. (2011). Comparative temperature tolerance in stingless bee species from tropical highlands and lowlands of Mexico and implications for their conservation (Hymenoptera: Apidae: Meliponini). Apidologie, 42(6), 679–689. https://doi.org/10.1007/S13592-011-0074-0/TABLES/2
- Maia-Silva, C., Silva Pereira, J. da, Freitas, B. M., & Hrncir, M. (2021). Don’t stay out too long! Thermal tolerance of the stingless bees Melipona subnitida decreases with increasing exposure time to elevated temperatures. Apidologie, 52, 218–229. https://doi.org/10.1007/S13592-020-00811-Z

Line 406 – This is an important note that, “the number of workers in a colony is important to consider.” Chelsea Cook has done some interesting research here about how the number of workers will influence the likelihood of fanning behaviors (Cook et al., 2016; Cook & Breed, 2013)
- Cook, C. N., & Breed, M. D. (2013). Social context influences the initiation and threshold of thermoregulatory behaviour in honeybees. Animal Behaviour, 86(2), 323–329. https://doi.org/10.1016/j.anbehav.2013.05.021
- Cook, C. N., Kaspar, R. E., Flaxman, S. M., & Breed, M. D. (2016). Rapidly changing environment modulates the thermoregulatory fanning response in honeybee groups. Animal Behaviour, 115, 237–243. https://doi.org/10.1016/j.anbehav.2016.03.014

Additional comments

Minor grammatical comments by line number:
Line 110 – need to specifically write out Bombus terrestris here since it is the first time this species is mentioned (not including the abstract).
Line 113 – Regarding the sentence, “This exposure to high temperatures may also affect the number of sexuals produced and their reproductive capacity (Guiraud et al., 2021), which is essential for reproduction and, consequently, colony and population success.” I would cut “reproduction and, consequently,” because it is redundant with the section of the sentence just before this part where you discuss reproductive capacity.
Line 122 – While it can be assumed that you are still discussing Bombus, technically, at no point did you explain that all bumblebees are within the genus Bombus. Because of that, I would either write out or specify somewhere that these species names are “Bombus …” since it is the first time you mention them in the paper.
Line 264 – depending on the formatting requirements of the journal, I would italicize these subheaders to make them clearer that they are nested underneath Colony Growth
Line 289 – There are suddenly no spaces between the numbers and symbols in this paragraph which is inconsistent with the rest of the paper.
Line 294 – I would specify in this subheader that it is “Drone Body Size and Dry Mass”
Line 414 – This is the first place where you mention the common name of this species. If you are going to do this, I would also mention it in the introduction.

Reviewer 3 ·

Basic reporting

The manuscript is well-written and clear. Literature cited is appropriate and Introduction is clear and well-cited. Data and code are available and the files all open.

One comment I have here is for the figures. In some figures the authors show significance or non-significance using letters, in other figures (including supp figures) this is not shown. I would request the authors make this consistent for all main text and supplementary figures as it makes the figures easier to follow.

Experimental design

All areas of this were generally well-done, however a few more details are required in the methods for replication.

1. Lines 198-212 - can authors be clearer about the timing information of heatwaves? If it took 30 mins to heat and 60 mins to cool to ambient, are these timings included in the overall four-hour heatwave period? I suspect not, but this needs to be very clear.

2. Perhaps describe in more detail how measures were taken. Many readers will not know how to take intertegular distance (and this occasionally varies). Also how weight change was measured could use more detail. Please check if the other measures have appropriate detail for replication as well. Other things such as % of syrup would be useful information.

Validity of the findings

Data and code have been provided. Conclusions were reasonable and discussed nicely in relation to literature.

Additional comments

Lines 258-262 - this section of the results needs to be tempered. The authors state that "Bumblebees exposed to the extreme heatwave exhibited higher mortality in relation to those from the control group (24 ºC) and the mild heatwave (32-34 ºC)." but go on to say the result was not statistically significant. The section should not start by indicated there was a significant change. The authors also state there is a trend with P = 0.22. A trend is generally considered as P < 0.1, so I would suggest removing this. Overall, this section claimed more than the data analyses showed.

I have no other comments and would recommend this study for publication after addressing my minor comments above. It is well written and a nice addition to the literature in this field.

---

## Round 0.2 · accepted · Accept

The authors seem to have addressed all comments raised by the reviewers; the manuscript is ready for publication.

·

Basic reporting

Confirmed

Experimental design

Confirmed

Validity of the findings

Confirmed

Additional comments

All above considerations are met. The questions raised in my review have all been satisfactorily addressed, and I think the manuscript is ready for publication.

·

Basic reporting

This paper provides a well-executed methodology addressing the effects of heat waves on bumblebee performance using microcolonies. The rationale, novelty, and impacts are well explained.
The authors thoroughly addressed all the reviewers' comments, even when some comments may have been difficult to address solely because of a lack of information available on certain topics (such as in-nest temperatures for this species during heat waves). I previously recommended major revisions because I had a few major comments that I felt needed to be considered before publication which the authors did a wonderful job addressing. They also thoughtfully and graciously addressed all the minor comments as well from the reviewers. I think the current form of this paper is in a good state for publication and have no further recommendations.

Experimental design

The experimental design was well thought out and took into account many aspects of colony performance for a well-rounded study that addressed the authors' objectives. This revised version of the manuscript addressed any confusion I had surrounding the specifics of what was done. I think the methods are well-described and thorough.

Validity of the findings

The findings of this research are interesting and valuable to the field. The authors did a good job of addressing all the reviewers' comments for this category of comments. I believe this manuscript is ready for publication, and I look forward to reading more research from the authors. I have no further comments.

Reviewer 3 ·

Basic reporting

n/a

Experimental design

n/a

Validity of the findings

n/a

Additional comments

The reviewers have made the changes I requested.